# Unexpected Emails to Submit Your Work: Spam or Legitimate Offers? The Implications for Novice English L2 Writers

**Josep Soler** *  **and Andrew Cooper**

Department of English, Stockholm University, Universitetsvägen 10E, S-106 91 Stockholm, Sweden; andrew.cooper@english.su.se

\* Correspondence: josep.soler@english.su.se; Tel.: +46-(0)8-163-593

**Abstract:** This article analyzes the discourse of what have been termed 'predatory publishers', with a corpus of emails sent to scholars by hitherto unknown publishers. Equipped with sociolinguistic and discourse analytic tools, we argue that the interpretation of these texts as spam or as legitimate messages may not be as straightforward an operation as one may initially believe. We suggest that English L2 scholars might potentially be more affected by publishers who engage in these email practices in several ways, which we identify and discuss. However, we argue that examining academic inequalities in scholarly publishing based exclusively on the native/non-native English speaker divide might not be sufficient, nor may it be enough to simply raise awareness about such publishers. Instead, we argue in favor of a more sociologically informed analysis of academic publishing, something that we see as a necessary first step if we wish to enhance more democratic means of access to key resources in publishing.

**Keywords:** academic publishing; predatory publishers; spam email; indexicality; linguistic repertoire; English L2 writers

---

## 1. Introduction

In recent years, interest towards what has been termed 'predatory publishing' [1] has grown considerably, with scholars from a range of different fields writing about the reasons for the emergence of this phenomenon, its implications for knowledge dissemination, and its impact on both science itself and scholars from around the world [2,3]. Discussions have mainly revolved around the danger that these publishers represent [4,5], not only because such journals appear to be content to publish virtually any paper that is submitted to them [6], but also because many of them seem to disregard any established ethical research standards [7]. For five years, between 2012 and 2017, Jeffrey Beall's blog Scholarly Open Access made a significant impact in raising awareness about disreputable publishers. However, beyond the sporadic cases that make it to the general media, such as Vamplew's emphatically worded paper [8], one of the most typical ways in which researchers have recently become acquainted with the phenomenon of predatory publishing is via the spam emails that companies from this type of business frequently engage with. This appears to be particularly true for those in the Arts and Humanities, disciplines that are still relatively unexploited by predatory publishers [2]; scholars in other fields, e.g., medicine and nursing, may have come across the phenomenon by reading the increasingly frequent warnings in editorial notes or full-fledged articles in established journals in their fields [9,10].

With this in mind, it seems important to conduct a discourse analysis of the primary means of communication that unknown publishers use in order to reach potential authors: 'spam' email.

Although spam is a recent phenomenon, previous work has already looked into the communicative forms and functions of such emails [11–14]. Our paper adds to the contribution of these initial studies, complementing the discussion from a sociolinguistic angle. Equipped with discourse analytic tools, we draw on the notions of indexicality [15] and the linguistic repertoire [16]. This is done in order to describe the typical features and intended functions of these emails, and to trace how certain meanings are accessed and activated (or not) through the use of certain linguistic and discursive forms, and how these resources are codified and potentially decodified by speakers (or readers/authors) with situated repertoires. The argument developed in the article is that the act of reading and decoding spam emails from predatory publishers as 'potential fraud' rather than 'legitimate offer' may not be as simple and straightforward as one might imagine.

In the article, we are particularly interested in addressing the following two questions: (1) what do these spam emails index? (2) are novice English L2 authors comparatively more likely to be affected by them? Anticipating our answer to these questions very briefly, we argue that emails sent from hitherto unknown publishers point to the very uneven playing field of present-day academic publishing, and that it is possible to imagine that English L2 writers, particularly novice ones, are indeed more likely to respond to their call for submissions, for reasons that have to do both with language competence and with the nature of one's situated position in the field of academic publishing. In a nutshell, we contend that authors' previous experience and their social position are key in shaping their capacity of interpreting and responding to spam emails. Together with both linguistic and communicative competence, authors' social position becomes a key element in determining the success (or lack thereof) of the communicative exchange generated by these spam emails. In summary, our analysis suggests further clues to understand why 'off-network' scholars may be more likely affected by publishers who engage in these email practices, in line with what previous research has already reported [2].

## 2. Background

Predatory publishing, a much ambiguous and debated term [17], is a recent phenomenon that has emerged from the conjunction of different factors, including the advent of open-access publishing and the increased pressure on authors from all over the world to publish more and more, and to publish 'internationally' [18]. Taking advantage of the 'gold open-access' model, in which authors pay author processing charges (APCs) in order to have their articles published, some companies seem to be ready to take scholarly publishing to its limits in order to exclusively generate revenue profits for their company. This kind of model builds on the commodification of science [19], in which quantifiable metrics are increasingly important in evaluating academics' scholarly performance. The system, thus, generates inherent pressures for scholars in need of quantifiable metrics and of publishing more and more in a larger variety of outlets. In this context, it is probably not surprising that it is younger, inexperienced scholars, from developing countries, that seem to have higher chances of publishing in predatory journals [3,20]. Previous studies have consistently reported that authors in predatory journals are based in Asia in their majority, with Africa in second place, whereas contributions from Europe and North America are in a minority [2].

In parallel to that, the field of English for Research and Publication Purposes (ERPP) has been growing in recent years, with one main question driving most of the work conducted in this area: how does the position of English as an international language of scientific dissemination affect the publication practices of scholars around the world? Previous work on ERPP has highlighted the pressures felt by multilingual scholars developing an international publication record [21]. It is argued that while English represents a useful tool for international cooperation, native speakers of the language enjoy an inherent advantage vis-à-vis non-native speakers [22]. More recently, authors have emphasized the idea that linguistic privilege exists in favor of native English-speaking academics, focusing on two of their privileges: for native English-speaking academics, publishing may: (1) require less effort (they might need to spend less time on their manuscripts), and (2) be biased in their favor (reviewers and editors will read their manuscripts more favorably by default) [23]. Thus, research on

ERPP can help to illuminate discussions on scholarly publishing more generally, with a focus on the challenges authors face in writing for publication purposes.

The reality of the native-speaker privilege for academic writers is, however, an open, ongoing debate in ERPP circles [24]. Indeed, language can be an important element in determining one's chances of getting published in top-ranking journals, but language alone does not seem to account for one's position in the field of academic publishing. In that respect, the term 'off network' has been proposed [25] in order to capture the 'center' vs. 'periphery' dichotomy [26] and to highlight the importance of the "need for access to networks of people who can facilitate the process [of publishing internationally]" [27] (p. 305). In the case of academic publishing, it is important to consider access to material and symbolic resources in defining the ways international publication spaces are shaped [28,29]. In sum, it can be argued that the native/non-native dichotomy is insufficient, and we should also be looking at scholars' means of production, their capacity to access well-resourced libraries, and their opportunities to engage meaningfully with relevant members of their discourse communities [30].

In order to understand the effectiveness of email communication that predatory publishers engage in, we need to say something more about two salient sociolinguistic concepts: indexicality and the linguistic repertoire. Indexicality is one key property of human language, referring to the connotational function of language, i.e., the capacity that language has to trigger symbolic, metapragmatic meanings [31]. Importantly, indexicality is not messy and unorganized, but it is in fact ordered and stratified: some "forms of semiosis are systematically perceived as valuable, others as less valuable, and some are not taken into account" [32]. The crucial point is one of access: access to producing and access to retrieving certain semiotic forms with a particular set of associated values. This is done not in a vacuum, but in real "contextualizing spaces in which particular forms can be attributed meaning" [31] (p. 76). This mapping activity can be problematic in two ways: on one hand, speakers may have different access to forms (resources), and on the other hand, speakers may have different access to contextual spaces, where forms are given value, where meanings are ratified or contested, and where forms are interpreted.

Crucially, therefore, we should emphasize the importance of the linguistic repertoires of those involved in any communicative exchange. In order to understand the effectiveness (or lack thereof) of texts like those we will analyze, we need to bear in mind the importance of the specific, situated repertoires, of those involved in the interaction, i.e., both sender and receiver's communicative repertoires. As individuals progress across their lifespan, their linguistic trajectory is inextricably linked and shaped by their life's path [16]. It is not only a question of different languages or language varieties becoming more present and more prominent in one's life, but it is actual linguistic features and knowledge of cultural norms that become more tangible. Thus, unequal life trajectories will almost necessarily result in unequal access to linguistic and cultural forms and to spaces where their functions are ratified, contested, or interpreted, where the indexical values of language forms are shaped.

## 3. Materials and Methods

We collected a corpus of 58 unsolicited emails offering publication services by hitherto unknown publishers. These emails were received by members of the Department of English at Stockholm University between March and October 2016. The corpus also includes emails received by the first author at his email account from a previous affiliation with the University of Tartu between August 2014 and October 2016. The emails included in our corpus were sent by companies that closely fulfilled the criteria that have been proposed to identify predatory publishers [4]. Considering these emails as a genre of their own, in conducting our analysis, we followed Askehave and Swales' 'bottom–up' approaches to genre description [33], which includes two directions. The first one is 'text-based', and the second one is more ethnographically oriented. In the text-based direction, the structure, style, content, and apparent purpose of some samples of the genre under study are examined and a description proposed [12]. In the ethnographic direction, the researcher takes a broader perspective, identifying

first the discourse community that engages in the communicative practices under examination and then analyzing their values, goals, and material conditions, rhythms of work, repertoires, and etiquettes, situating the genre within those contexts [34].

In our study, we adopted both strategies, with a line-by-line reading of the texts, comparing and contrasting their rhetorical moves, and placing them in their broader sociopolitical context, with an interpretation of their possible broader meanings. These two directions of text analysis usefully highlight two necessary dimensions of text-based research: the micro level of fine-grained details of texts and the macro level of their position within the communicative practices of a discourse community. In the section below, we present the 'bottom–up' or the fine-grained analysis of the email texts that we collected and we place them in their broader context, the macro level, in the discussion section below.

## 4. Results

Emails from predatory publishers present similar structures and rhetorical moves [14], with one implicit purpose: to convince their recipients to submit original research papers. The emails in our corpus can be divided into two types: (1) emails with an introductory letter containing a prototypical email text, with a salutation line and two or three short paragraphs explaining why the author is being contacted and giving some brief details about the journal, with a concluding greeting note and signature of the sender, and (2) emails with a longer description of the journal, normally under the heading 'call for papers' with instructions for authors, review policy, etc. Type 2 emails are less personal and they typically do not contain a reason explaining why the recipient is being contacted, but they might end with a salutation and signature by the editor in chief or an editorial assistant. Detailed descriptions of the submission and review procedure are sometimes included. Table 1 summarizes the prototypical rhetorical moves in Type 1 emails, together with examples from our corpus.

Much like in Blommaert and Omoniyi's [35] analysis of financial email scams, emails in our corpus showed a gradation of competences and skills; from good/very good IT literacy skills, to fair knowledge of email communication conventions, to poor/very poor communicative and linguistic competence. As we summarize in Table 2, the level of IT skills of email authors seems undeniable, with a capacity to generate personalized emails beyond the salutation line to potentially a very large number of recipients. Senders sometimes use different servers and generic free email providers (e.g., Yahoo, Hotmail) in addition to employing concealed or fake message senders and signers [11]. Indeed, those emails following letter format conventions are 'signed' in the sense that they feature a name and contact details after the main body text. In no instance in our corpus, however, does the email sender's name match with the signatory in the email text. This might indicate an awareness of the importance of remaining anonymous by the senders, as previous research has already noted [11,35].

Whenever they claim an explicit location, most journals in our corpus claim to be located in the USA, a piece of information that is sometimes explicitly repeated in the email. Some give several locations, in which case the USA is always included. The other countries that are explicitly mentioned in our corpus are China, the UK, India, the Netherlands, Germany, and Indonesia. In some, two addresses are given—a principal place of business in China and a post office box in the USA. This indicates that for the majority of the companies in our study, presenting themselves as associated with the USA is an important element to include in the email.

**Table 1.** Rhetorical moves of contact emails from predatory journals, with examples from our data.

| Move | Example |
|---|---|
| 1. A personalized salutation to the author | Dear ((Last name, First name))<br>Dear ((FIRST NAME LAST NAME))<br>Dear Ms/Mr ((Last name, First name))<br>Dear and ((First name, Last name)) |
| 2. A brief introduction of the journal, with its title name, and an explicit reference to its quality attributes. | This is *Journal of [Anonymized]* (ISSN 21XX-79XX), a professional journal published worldwide by Academic [Anonymized] Company, New York, NY, USA. |
| 3. A reference to a paper recently presented by the author at a conference that the journal would be interested in publishing. | We have learned your paper "((Paper title))" at ((Name of Conference)). We are very interested to publish your latest paper in the Journal of [Anonymized]. |
| 4. A note of praise to the author. | Through your works, I know you are an expert in this field. We are seeking submissions for the forthcoming issue published in December 2014 currently.<br>Your submission will make an important contribution to the quality of this journal. |
| 5. An invitation for the author to submit any further unpublished material that they may have. | All your original and unpublished papers are welcome. |
| 6. An invitation for the author to become a reviewer or editorial member of the journal. | We are also recruiting reviewers for the journal. If you are interested to be a reviewer, it's our great honor to invite you to join us. . . . After assessment, the editorial board will decide whether we offer you the position of reviewer or not. |
| 7. An appeal to the author to stay in touch and the establish a stable relationship. | Hope to keep in touch by email and publish some papers or books from you and your friends in USA. As an American academic publishing group, we wish to become your friends if necessary. Expect to get your reply soon. |
| 8. A concluding salutation, and signature of the sender, including their contact details | Best regards,<br>Cindy<br>Journal of [Anonymized]<br>Academic [Anonymized] Company<br>jmer@[anonymized].us,<br>education@[anonymized].us<br>228 East 45th Street, Ground Floor,<br>#CN00000267 New York, NY 10017, USA<br>Tel.: +347-XXX-2153, +347-XXX-6798,<br>Fax: +646-XXX-4168, +347-XXX-1986 |

**Table 2.** Examples of senders' names, aliases, and addresses in our corpus.

| | |
|---|---|
| Concealed or fake email signer | Dr. Michael King; Laura; Ivy; Cindy; Cassiel; Sunny, H.; Lily, R. |
| Concealed or fake email sender | Vladimir Ushmov; Jeniffa Lopez; Rosariar Caleca; Rearden Mamie; Vensate Henry; Bertha Louis; |
| Concealed or fake publisher's addresses | Principal Place of Business: Building 5, Headquarters Space of Optical Valley, Tangxun Lake North Road #38, East Lake High-Tech Development Zone, Wuhan 430223, Hubei Province, China.<br>For ease of communication: Scientific Research Publishing Inc., P.O. BOX 54821, Irvine, CA 92619-4821, USA. |

As already mentioned, the key purpose of these emails is to convince the recipients to submit their work to the advertised journal. To that end, senders mobilize two strategies: praising the recipient and constructing them as an expert in the field (regardless of how misplaced or exaggerated that praise may be) and, at the same time, building up an image of the journal and the publisher as a rigorous and professional one. Moves 4 and 6 in Table 1 above illustrate these two strategies. Sometimes, however, senders may use subtler ways of trying to entice a response from the recipient, as in the example

below, where previous (non-existent) email communication is referred to. As well as constructing the recipient as a busy person who may not notice correspondence, this also activates a sense of guilt over uncompleted tasks, which may encourage a quick response by the reader:

> Did you get my last email about Invitation for paper submission?
> We hereby forward it again, please kindly confirm receipt.
> If you or your colleagues (or friends) have any unpublished papers, please kindly submit it by email attachments.
> Looking forward to hearing from you soon.

In parallel to that, spam emails from predatory publishers are replete with semiotic devices that aim at constructing the sender as a legitimate, reputable, and quality publisher. These may include: An indication of the impact factor of the journal (usually self-calculated), a list of databases in which it is indexed, a profuse repetition of ISSN numbers in the text, large chunks of numbers and other signs in the contact details of the sender, etc. Indeed, we see in these emails a collection of seemingly meaningless semiotic signs. Much like with the more general financial fraud emails analyzed by Blommaert and Omoniyi [35], these features do not seem to have any referential value, but they may work at an indexical level, in their attempt to convey an image of a well-established, professional, and trustworthy company.

*Problematic Features in Predatory Publishers' Emails: Micro-Level, Linguistic Details*

As already seen from the examples above, the texts in the emails of our corpus exhibit a number of instances of odd phraseology, combined with features of grammatical, lexical, and syntactic nature that differ from standard English. It is precisely at this level of communicative and linguistic competence in which the more fine-grained details become salient and may unmask the connection between the senders of these emails and their claimed identities as professional and trustworthy publishers. Herley [36] has proposed that in more general fraud scams, nonstandard features serve as a filter to dissuade those who are not likely to go through the scamming process while attracting those who are more likely to do so. In the academic publishing emails that we analyze, this seems less feasible, since the very nature of communication in the field requires more control of the standard and of specific formulae. Even so, nonstandard features are apparent in many instances of the email texts, sometimes hard to notice, sometimes with highly marked choices, denoting the "unfinished" [37] character of the repertoire of their authors (marked choices understood in this analysis as variations from standard written English).

One aspect that authors of the emails in our corpus control at a fairly successful level is that of punctuation and capitalization issues, or "grassroots literacy" [35]. There is a very limited number of problematic instances at that level, although sometimes, problems of this type can appear in key moments of the email, as in the opening salutation line (see the instances above in Move 1 in Table 1 above, with marked choices of capitalization norms, as well as choice of title of address, and the order of the name and last name of the addressee). Other examples of marked choices in the opening salutation lines of the emails include nonpersonalized salutations, which may include long streams of titles and ranks, or typos that are very revealing. Signature lines can also include marked choices, which can range from less to more noticeable. For example, in the emails sent by one specific company, the text is signed invariably by "Best regards" followed by only the first name of the email author, without the last name; in professional emails of this type, signing without the last name in a first-contact message is uncommon. In other occasions, signature lines are more marked than that, as we can see in some of the examples from our corpus included in Table 3.

**Table 3.** Examples of fine-grained lexical and grammatical issues, and odd formulations.

| Odd Phrasing or Word Choice | Example from Our Corpus |
|---|---|
| Salutation line | "Esteemed Sir/Madam"" Dear Professors/Teachers/Academicians/Research Scholars" "Dear Scholar/Processor" |
| Concluding line | > Thank you, > With best regards > Looking forward for long lasting academic relationship > With Regards |
| Aims of the journal | The organization aims at undertaking, co-coordinating and promoting research and development. It provides professional and academic guidance in the fields of basic inculcation, Higher inculcation. Engineering Research Publication mission is to Promote and support, High Quality basic, Scientific Research and development in the fields of Engineering, Technology and Sciences. Generate Public awareness, provide advice to scholar's researchers and communicate research outcomes. |
| Advertisement: Special offer | Grasp the Privilege for Publishing Paper . . . Only in 30–50 days, paper with good quality can be published. |
| Enticing authors to submit their work | > We do not only published papers, but also spread them to other channels to increase their downloads and citations. We also help submit papers to databases and indexes. We have been doing our best for world wide researchers, since we believe science is fantastic and your research is fantastic. > We honestly ask you to submit paper to our newly launched journals. ( . . . ) You can publish one paper without APC in one of the our newly launched journals if you submit the paper before 31 October 2016. Here is some newly launched journals. |

In many cases, a superficial reading of the emails might not lead to raising serious suspicions about the texts. However, close reading of the material reveals frequent grammatical and lexical errors of higher or lower intensity. The most easily identifiable of these are collocation and word choice errors. Passages which aim at emulating a formal academic English reflect marked problems of word choice (e.g., *inculcation* for *education*), capitalization errors, and coordination errors. Odd word choice is a common feature in much of the corpus, which also includes word order errors (*only in* instead of *in only*), problems of noun inflection, and sentences without main clauses.

Convincing scholars to submit their work to the journal is probably one of the more delicate moves in the email. Ironically, it is oftentimes precisely in this part of the text where more marked and nonstandard choices are apparent; many times, such choices are apparent in oddly formulated passages. Not infrequently, in those emails comprising running text, attempts to persuade researchers go beyond odd formulations and also include marked errors of syntactic and grammatical nature, as in the extracts in Table 3 above. All in all, the messages in our corpus contain details of a grammatical, stylistic, and pragmatic nature which are problematic and that distance their authors from the image of the professional and reliable academic partner that they are trying to convey. We turn next to discussing the meanings of the emails in our corpus and conclude with a more general reflection on the nature of predatory publishers' emails in the context of present-day academic publishing.

## 5. Discussion and Conclusions

We have seen above that emails from predatory publishers present a gradation of competences by their authors in different areas: from good/very good skills in IT and computer communication to fair knowledge of email communication, and poor/very poor pragmatic and linguistic competence. The latter two are what appear to be the more problematic ones; indeed, it is at that level that the emails

may become eventually decodable as 'potential fraud', rather than 'reliable publisher'. Here, however, we need to return to and further develop the main argument we wished to present in the article: that retrieving one meaning rather than the other one, reading these messages as fraudulent rather than legitimate, may not be as straightforward as one can anticipate, and potential interpretations are contingent on speakers/readers' past experiences and situated linguistic repertoires. The concept of indexicality is key to understanding how the interpretation process of these email texts is developed, the idea that "every emblem of distinction in societies is subject to ... the delicate play of voices in polyphonic discourse ... of availability and accessibility, inclusion and exclusion" [32] (p. 128). Thus, in accounting for what happens in the communication chain of these emails, we need to place significant emphasis on the individual speakers (writers or readers) involved in it, with their situated linguistic repertoires.

We cannot simply take it for granted that because certain linguistic features in these texts are obviously problematic to some readers, these features will invariably index 'potential fraud' instead of 'reliable publisher', counter to what is commonly argued (more tacitly than overtly) in discussions of predatory publishers [38,39]. Activating one or the other kind of interpretation depends on the reader's socialization trajectory and the opportunities they may have had to access social spaces where the meanings of specific linguistic and cultural features are constructed. The key here is the availability of the 'right kind' of linguistic features and the accessibility of the 'right kind' of cultural and social spaces where the values of these features are validated, negotiated, or challenged [31]. Seen from this perspective, we can better grasp and fine-tune the claims by previous authors that it is particularly junior colleagues in developing countries who are more likely to send their work to predatory journals [3], and (we may add) who are also more likely to be English L2 writers. It may or may not be the case that the publishers specifically target these groups, but we would contend that in more than just a few cases, these researchers may have had life trajectories which put them in a limiting position to identify these emails as coming from 'fraudulent' instead of 'genuine' publishers. Indeed, as noted for more general email scams [37], messages like these do work insofar as the communicative resources employed in the production of the text and the repertoire of the text producer overlap to some extent with those of the receiver of the text. Briefly put, addressing the first of our research questions, we suggest that the sheer circulation of these spam emails indexes and points at the very crude inequalities that exist in the field of present-day academic publishing.

As for our second question, we asked ourselves if these emails and the publishers that engage in these practices may more intensely affect English L2 scholars. In response to that question, we suggest that the discourse analysis presented above does indicate that, indeed, novice English L2 authors may be more affected by such publishers. In line with the above argument in connection to the inequalities present in today's field of academic publishing, spam email invitations have the potential to show global disparities, whereby unequal forms of English [40] point to unequal forms of access to linguistic, social, and cultural capital. In that sense, we would argue that English L2 writers can be doubly affected by the issue of predatory publishing. At a fundamental level, some of these authors might more easily overlook the fine-grained linguistic details that indicate the potentially illegitimate nature of such email offers. At a deeper level, beyond issues of purely linguistic competence, other authors who submit their work to such journals are not simply victims that fall prey to these publishers. The meagre empirical research that exists so far on the question of why scholars decide to submit their work to these journals points to reasons that have to do with authors' self-perceived notion of their position in the field of academic publishing. In short, contributors to predatory publishers report a perceived bias against them as authors from developing countries [41,42], so that the route to publishing becomes easier and more straightforward with these outlets [18,43].

The above discussion may allow us to fine-grain some of the ongoing debates on the question of English for Research and Publication Purposes (ERPP) and on scholarly publishing more generally. On the question of ERPP, and whether English represents an additional barrier that second-language writers need to overcome, we would argue that framing the discussion in terms of injustices and

inequalities between native and non-native English-speaking scholars is important [23], but perhaps not sufficient [24]. Language can certainly help us to point at the inequalities existing in the field of academic publishing, as our analysis has attempted to show, but language alone does not explain the existence and the perpetuation of such inequalities, as discussed above. Indeed, predatory publishing is a complex phenomenon. Indeed, even if less intensely, English L1 scholars may be affected by it and may as well submit their work to predatory journals. Pyne's study is particularly revealing in that sense, showing that publications in such journals do not impinge negatively on academics' career prospects at a small business school in Canada. The author explains that "at least at one university [his case study], there are few incentives not to publish in predatory journals" [44] (p. 156). Predatory publishing, therefore, escapes easy and straightforward readings, much like questions on the status and the role of English in perpetuating linguistic injustices in the field of academic publishing.

To conclude, we certainly need more studies that investigate the motives and the experiences of authors publishing in predatory journals in more detail [41,42]. At this point, even if we concede that some L1 English authors from more 'central' positions might be also affected by them, it seems proven that the majority of authors in these journals are younger and less experienced academics from developing countries [2,3,20,45], and it is probably not a coincidence that this is the case. Calls for raising awareness of fraudulent practices associated with open-access publishing in these contexts are definitely important [43], and these calls may help some researchers to avoid the traps of these publishers, but that alone will not be enough. We may continue asking: why is it that some researchers overlook the poorly formulated email texts by predatory publishers, or even why do some ignore the lack of proper peer-review processes and pursue with publication in these journals anyway (thus colluding with fraudulent publishing)? [46]. The key point, once again, is that different socialization opportunities can lead to different ways of approaching the publication process, and at least for some, the deceptive nature of predatory publishers will remain difficult to uncover. Addressing structural inequalities and thinking of more inclusive and radically democratic ways of scientific dissemination seem necessary alternatives to explore.

**Author Contributions:** Conceptualization: A.C. and J.S.; formal analysis J.S. and A.C.; writing—original draft preparation, J.S., review and editing, A.C.

**Funding:** This research received no external funding.

**Acknowledgments:** We wish to thank our colleagues Maria Kuteeva and Kathrin Kaufhold at the Department of English of Stockholm University for constructive feedback and close reading of an earlier draft of this paper. An oral version of it was presented at the English Higher Seminar of Stockholm University (March 2017); at the conference "How does vulnerability matter" in Helsinki (December 2017); and at PRISEAL-4 Conference in Reykjavik (September 2018). We wish to thank colleagues who commented on the paper and provided useful insights at these several occasions. A working draft of the paper appeared at the *Tilburg Papers in Culture Studies* (Paper 184); we thank Dr. Massimiliano Spotti and colleagues in Tilburg for their support. All remaining errors and shortcomings are, naturally, our own.

**Conflicts of Interest:** The authors declare no conflict of interest.

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
