# Peer review of "Unexpected Emails to Submit Your Work: Spam or Legitimate Offers? The Implications for Novice English L2 Writers"

_publications, doi:10.3390/publications7010007_

Reviewer 1 Report

I enjoyed reading this article. It is original and of current interest. It does satisfy the journal purposes.

The introduction guides the audience into the topic smoothly and with examples and previous studies. The explicitness of the research questions at the end of the introduction provide the reader with the arguments to follow along the paper.  The research question one in a way addresses the first part of the article’s title and the second RQ, focuses on implications for L2 writers. I believe this was strategically done. Good to see this.

The literature is logically organised and discussed providing solid basis for the construction of the arguments. It does intrigue the readers with previous research and engages them into the present one.

Regarding the Materials and methods / these are clearly presented. But perhaps line 129 – 130 clarify why/how the emails were refined to 25 types. Or mention they will address this in a specific section.

Line 149 “In this section” the current materials and method section that is finishing with that section?

The layout of tables with the item to analyse and then the example next to it makes the results clear.

Well done. It does answer he RQs and points the implications for L2 writers and the awareness of the issue.

Author Response

We thank the reviewer for their very useful comments and feedback. Please, find attached a document with our detailed answers to their points.

Reviewer 2 Report

Review

Summary

I believe the topic is of interest to the journal’s readership. The paper reports on

the analysis of a sample of spam emails received from unknown and potentially predatory journals. The principal argument advanced by the authors is that discerning predatory emails is

not easy, especially for L2 and peripheral scholars who – compared to their L1 peers in the Western academic centre – are more likely to lack the requisite linguistic repertoire and experience to be able to distinguish between genuine and predatory journals. The authors also report that they found a marked heterogeneity among the (predatory) emails in their study with respect to the level of linguistic competence/skills that they showcase.

The authors have done a good job of locating their study within the recent emergent scholarship on the phenomenon of predatory publishing. Overall, the piece is a well presented and readable paper, and will be of interest to the readership of Publications. However, there are some revisions that need to be addressed before the paper can be considered for publication.

# Specific comments:

Title:

I suggest adding the term “peripheral” or “novice” to “English L2 writers”.

The authors in fact argue that identifying fraudulent solicitations to publish depends not only on one’s linguistic repertoire but also on their “previous experience and their social position are

key in shaping their capacity of interpreting and responding to spam emails … [and] … ‘off-network’ scholars may be more likely affected by publishers who engage in these email practices.” (p. 2) The same argument has been reiterated on page 8 (Lines 276-280). This contention needs to be reflected in the title as well.

Abstract:

The authors argue that it may not “be enough to simply raise awareness against such publishers.” (p. 1) I am not sure if this point is referred to/elucidated in the discussion/conclusions.

Background:

Lines 125-127

The authors state that “to gain a full perspective of the ‘total linguistic fact’ [15] generated by the emails like those we shall analyze, we need to understand their indexical field and indexical values, and the resources and repertoires available to all of those involved in such kind of email communication, the senders and the receivers.” (p.3)

I am not sure how/whether the data presented here would elucidate the above points.

Materials and Methods:

Lines 129-130

Explicating the methodology, the authors  report that they “collected a corpus of 58 tokens of unsolicited emails offering publication services (later refined to 25 types)”. The latter is not referred to/discussed in the paper.

Results:

Did the data analysis entail a tally of the frequencies and proportions ? If so, they can be easily visually displayed. Doing so can help offer a clearer picture of which features occurred more frequently than others in the analyzed emails.

Rather than using unspecific quantifiers/adverbs – e.g., “most emails” (Line 173), “plenty of”(Line 253) and “Not infrequently” (Line 251) – the authors could present the frequencies and proportions of the discerned features in the analyzed emails

Line 176

The statement “This indicates an awareness of the importance of remaining anonymous by the senders” needs some qualification and hedging.

Lines 214, 222, 250 

In describing the problematic features of the analyzed emails, the authors have used the term “deviant” four times. I suggest using less loaded/strong terms such as “variations from standard in English” or “ linguistic/grammatical inaccuracies”. 

Lines 209-210

The strong claim that “these features do not have any referential value at all, but they completely work at an indexical level, in their attempt to convey an image of a well-established, professional, and trustworthy company” needs to be modulated and nuanced.

Lines 223, 229, 230

It’d be better to unpack the term “un/marked choices”, at least in a footnote, so that it is clear enough to the reader. (pp. 6-7)

Line 241

The authors state “In many cases, a superficial reading of the emails might not lead to raising serious suspicions about  the  intentions  of  the  senders.” (p. 8)

How can we determine the intentions of the sender? Do linguistic inaccuracies/infelicities in an email necessarily indicate the predatoriness of a journal?

Discussion and conclusions:

In general, a more in-depth interpretation/discussion (of the findings) with reference to the

notion of “indexicality” – which underpins the conceptual framework of the study –

is warranted.

Line 260

What is exactly meant by “IT literacy skills”? Was this part of the research questions of the study?  (p. 8)

Line 277

What is precisely meant by “reader’s trajectory”. Trajectory in terms of academic advancement, publication records? Please specify.

Line 334

The authors state “different socialization opportunities will lead to different ways of approaching the publication process” (p. 9).

I suggest replacing “will” – which conveys certainty – with “often” or “can” to qualify it.

References:

The following references do not seem to be accurate and need to be double checked:

7. Bohannon, J. Who’s Afraid of Peer Review, Science 2013 342, 60-65.

 Question mark is missing.

36. Herley, C. Why do Nigerian scammers say they are from Nigeria? Microsoft website. Available online from:

https://www.microsoft.com/en-us/research/publication/why-do-nigerian-scammers-say-they-are-from-nigeria/ (accessed 14 November 2018)

42. Shehata, A.M.K.; Elgllab, M.F.M. Where Arab social science and humanities scholars choose to publish: Falling in the predatory journals trap Ahmed, Learned Pub. 2018 31, 222-229.

# Minor (language/style) issues:

On a different note, there are some minor local issues, which need to be addressed. Here, for the purpose of illustration, are some examples but this is not a comprehensive list:

Line 49

“do these spam emails index? And (2) are English L2 authors” (p. 2)

Lines 159-160

“Emails of ‘call for papers’ only type are less personal, they typically do not contain a reason explaining why the recipient is being contacted…” (p. 4)

Lines 263-264

 “Here, however, we need to return and develop further the main argument” (p. 8)

The preposition “to” is missing.

Line 286

“limiting position to identify these emails as ‘fraudulent’ instead of ‘genuine’ publisher.” (p. 9)

Line 294

“that indeed, English L2 authors…” (p. 9)

The comma is redundant here.

Line 302

“victims that fall prey of these…” (p. 9)

preposition usage –“to” rather than “of”

Line 324

“… even if we may concede that some” (p. 9)

The word “may” seems redundant.

Line 328

“practices associated to open-access publishing” (p. 9)

 preposition usage –“with” rather than “to”

Line 330

This sentence contains an embedded question and needs to be fixed.

“We may continue wondering why is it that some researchers overlook the poorly formulated email texts by predatory publishers” (p. 9)

Author Response

(The authors gave the same response as above.)
